# Assessment of the Performance of Ultrasonography for Detecting Myofascial Trigger Points

**DOI:** 10.3390/s24030718

**Published:** 2024-01-23

**Authors:** Han-Yu Chen, Chang-Zern Hong, Yueh-Ling Hsieh

**Affiliations:** 1Department of Physical Therapy, Hungkuang University, Taichung 433304, Taiwan; hychen99@hk.edu.tw; 2Taiwan Myopain Academic Association, Tainan 704032, Taiwan; johnczhong.john@gmail.com; 3Department of Physical Therapy, China Medical University, Taichung 406040, Taiwan

**Keywords:** ultrasonography, myofascial trigger point, endplate noise, echogenicity

## Abstract

Needle electromyogram (EMG) research has suggested that endplate noise (EPN) is a characteristic of myofascial trigger points (MTrPs). Although several studies have observed MTrPs through ultrasonography, whether they are hyperechoic or hypoechoic in ultrasound images is still controversial. Therefore, this study determined the echogenicity of MTrP ultrasonography. In stage 1, the MTrP of rat masseter muscle was identified through palpation and marked. Needle EMG was performed to detect the presence of EPN. When EPN was detected, ultrasound scans and indwelling needles were used to identify the nodule with a different grayscale relative to that of its surrounding tissue, and the echogenicity of the identified MTrP was determined. In stage 2, these steps were reversed. An ultrasound scan was performed to detect the nodule at the marked site, and an EMG needle was inserted into the nodule to detect EPN. There were 178 recordings in each stage, obtained from 45 rats. The stage 1 results indicate that the MTrPs in ultrasound images were hypoechoic with a 100% sensitivity of assessment. In stage 2, the accuracy and precision of MTrP detection through ultrasonography were 89.9% and 89.2%, respectively. The results indicate that ultrasonography produces highly accurate and precise MTrP detection results.

## 1. Introduction

Myofascial pain is a common cause of temporomandibular disorders (TMDs) [1]. Therefore, myofascial pain of chewing-related muscles should be treated before myogenic TMDs are managed. Myofascial pain is a common musculoskeletal condition characterized by the formation of myofascial trigger points (MTrPs), which are hyperirritable nodules that form in the taut bands of muscles [2,3]. Clinicians usually diagnose myofascial pain syndrome by identifying one or more MTrPs [4]. Previously, they often palpated a taut band and used the resulting referred pain as the basis for determining the location of an MTrP. This physical examination for locating MTrPs has been widely used [4,5]. However, numerous scholars have suggested that this method is unreliable [6].

In electrophysiological studies, spontaneous electrical activities, including endplate noise (EPN) and endplate spikes, were detected after a needle was inserted into an MTrP [7]. Additionally, the incidence of EPN is significantly higher at MTrPs than at non-MTrP sites [8], and EPN is positively correlated with MTrP irritability [9]. Consequently, scholars have asserted that EPN is a characteristic manifestation of MTrP that is helpful for diagnosing TMDs [8]; notably, it is not an absolute diagnostic standard. Although needle electromyography (EMG) can be used to identify MTrPs in a research setting, it is an impractical option in a general clinical setting [10].

Ultrasonography is also used to diagnose MTrP because it is straightforward, cost efficient, and radiation free. Elastography is a method for quantifying the viscoelasticity of soft tissue, which is achieved by observing the tissue’s response to external stress or vibration. Because an MTrP is stiffer than its surrounding tissue, its location can be determined on the basis of this characteristic [11,12,13]. Although studies have extensively demonstrated the application of ultrasonography for detecting MTrPs, this method requires additional induction and is still impractical for general clinical settings. By contrast, a more suitable clinical practice is to directly observe MTrP as two-dimensional (2D) images and to determine their location by differentiating the echo characteristics of MTrPs and their surrounding tissue; notably, the basis for image interpretation in this context remains controversial. Several studies have suggested that the ultrasound images of MTrPs exhibit hypoechogenicity [13,14]; however, others have reported that such images exhibit hyperechogenicity [15,16]. Hypoechogenicity occurs when an ultrasound produces less echogenicity upon encountering a tissue or substance and presents a darker image of the tissue or substance site relative to its surrounding tissues. For example, liquids such as urine are hypoechoic, whereas hard tissues such as bones are hyperechoic [17]. Therefore, this study examined the occurrence of EPN to determine the ultrasonography imaging characteristics of MTrPs and assess the performance of ultrasonography for detecting MTrPs.

## 2. Materials and Methods

### 2.1. Animal Care and Preparation

Experiments were performed on adult male Sprague–Dawley (SD) rats (250 to 300 g), purchased from BioLASCO, Taiwan. The animals were kept on an artificial 12 h light–dark cycle at a university animal center. Food and water were available ad libitum. Each animal was housed and cared for in accordance with the ethical guidelines of the International Association for the Study of Pain [18,19]. Effort was made to minimize the discomfort of the animals and to reduce the number of animals used. All animal experiments were conducted with the procedure approved by the Animal Care and Use Committee of the university, in accordance with the Guidelines for Animal Experimentation (No. 2018-059).

### 2.2. Identification of MTrPs

A specific hyperirritable spot (MTrP) in the masseter muscle of rats is similar to that located in the masseter muscle of humans. At this spot, local twitch responses can be elicited when a needle tip encounters a sensitive locus [20]. Similarly to human MTrPs, this sensitive spot of rats frequently exhibits spontaneous electrical activity (e.g., EPN) [8,21]. Before an anesthetic was administered, the most tender spots (i.e., MTrPs) of randomly selected masseter muscles were identified via finger pinching. An animal’s reaction (e.g., withdrawal of the lower limb, head turning, and screaming) was observed to confirm the exact location of an MTrP [22,23,24]. These painful regions were marked on the skin with an indelible marker, and the animals were anesthetized with isoflurane (AErrane, Baxter Healthcare of Puerto Rico, PR, USA) in oxygen flow (2% for induction and 0.5% for maintenance) [25]. Body temperature, which was monitored by inserting the thermistor probe of a thermometer (Physiotemp Instrument, Clifton, NJ, USA) into the rectum, was maintained at approximately 37.5 °C; this was achieved using a body temperature control system with a thermostatically regulated DC current heating pad and an infrared lamp. The masseter of the marked side of the face of a rat was held between the fingers from behind the muscle, and the muscle was palpated by gently rubbing (rolling) it between the fingers to identify a taut band. A taut band feels like a clearly delineated rope of muscle fibers (with a diameter of approximately 2–3 mm) to the touch. Such areas were designated for the evaluation of ultrasound images and electrophysiological recordings.

### 2.3. Electrophysiological Recording of EPN

For EPN assessment, a two-channel digital EMG machine (Neuro-MEP-Micro; Neurosoft, 5, Voronin Str, Ivanovo, Russia) and monopolar needle electrodes (37 mm disposable Teflon-coated model) were used. The gain was set to 20 μV per division for the recordings from both channels. Low-cut and high-cut frequency filters were set to 100 and 1000 Hz, respectively. Sweep speed was 10 ms per division. The search needle used for the EPN recording was inserted into the MTrP region and connected to the first channel of the EMG machine. The control needle was inserted into the same muscle in the non-taut band region near the MTrP and connected to the second channel. The common reference needle electrode for each channel was secured on the skin through an incision and connected to both channels through a y-connector.

The search needle was inserted into the MTrP region parallel to the muscle fibers at an approximate 60° angle to the surface of the muscle. After its initial insertion to a depth just short of an MTrP or to a comparable depth at a control site, the needle was advanced very slowly, being simultaneously and slowly rotated in order to prevent it from suddenly grabbing and releasing tissue to advance with a large jump. Each advance covered only a minimal distance (approximately 1 mm). When the needle approached an active locus (EPN locus), a continuous distant electrical activity (i.e., EPN) was detected. When EPN with an amplitude of >10 μV was recorded, the examiner stopped advancing the needle and gently moved the needle minimally in a different direction; this change was made to obtain the EPN with the highest amplitude. If the desired result was not achieved, the needle was advanced to another site until an EPN with an optimal amplitude (usually >30 μV) was recorded. Subsequently, the needle was fixed in place (carefully and firmly taped onto the skin) to ensure the continuous run of the EPN on the recording screen at a constant amplitude. Throughout the experiment, continuous EPN tracing was performed so that EPN changes were continuously visualized on an EMG screen (Figure 1). If the EPN was unsustainable, the searching needle was moved to another site until a satisfactory EPN tracing result was obtained. This EPN recording procedure was performed by an investigator who was blinded to the group assignment.

### 2.4. Ultrasound Imaging

Morphological data on the MTrPs of masticatory muscles were mainly acquired through ultrasonography (Terason t3000 Ultrasound System, Ormond Beach, FL, USA). All ultrasonography evaluations performed in this study were performed by a single assessor who was familiar with ultrasonography operation and interpretation and was blinded to the grouping status of the tested animals. After the location of an MTrP on a rat was determined through palpation, a 7–12 MHz linear array transducer (Terason 12HL7, 25 mm hockey stick style, 128 elements) was placed on the masseter muscle parallel to its upper jaw with minimal pressure applied. Ultrasonography was conducted at a focal length range of 0.3–1.0 cm and an image depth of 1 cm. In our pilot study, the EPN locations detected using the EMG needle were scanned by performing an ultrasound, and the identified muscle nodules were mostly revealed to be hypoechoic (Figure 2). Therefore, the assessor determined that the hypoechoic nodules were MTrPs, captured the relevant images, and recorded them for further analysis.

### 2.5. Experimental Procedures

On the basis of the research purpose, the present experiment was divided into two stages. Stage 1 was performed to confirm whether the identified MTrPs were hypoechoic or hyperechoic. The assessor inserted the needle into the masseter muscle and moved it slowly depending on the signals that they observed. During EPN detection, an ultrasound scan was performed with an indwelling needle to locate the MTrP. The nodule with a grayscale that differed from that of its surrounding tissue was identified, and its image was captured and recorded for further analysis. In stage 2, the steps performed in stage 1 were reversed in order to assess the performance of ultrasonography for detecting MTrPs. Ultrasound images of MTrPs were captured and recorded on the basis of the characteristics as determined in stage 1. A needle was inserted into the MTrP site as identified under ultrasound guidance in order to record the electrical signal of the site.

### 2.6. Data Analysis

The parameter data obtained in this study exhibited both “positive” and “negative” when referring to the presence and absence of the phenomenon, respectively. In stage 1, the phenomenon indicated the presence of hypoechoic images on the site of the exhibited EPN. And the phenomenon indicated the presence of EPN on the site of hypoechogenicity in stage 2. All data were compiled in a crosstab, and their sensitivity, specificity, positive predictivity, negative predictivity, positive likelihood ratio, negative likelihood ratio, accuracy, and precision were calculated.

## 3. Results

A total of 45 rats were included. Bilateral masseter muscles of each rat were measured twice in each stage. However, two recordings were discarded due to poor quality. Therefore, there were 178 recordings for each stage. In stage 1, images indicating hypoechogenicity were obtained for the 148 MTrPs that exhibited EPN. Of the 30 MTrPs that did not exhibit EPN, images indicating nonhypoechogenicity and hypoechogenicity were obtained for twenty-eight and two MTrPs, respectively. In stage 2, among the 166 MTrPs for which hypoechoic images were obtained, 148 exhibited EPN, whereas 18 did not do so. The 12 MTrPs for which hypoechoic images were not obtained did not exhibit any EPN.

The data from stages 1 and 2 are listed in Table 1 and Table 2, respectively. The results pertaining to the performance of ultrasonography for detecting MTrPs are summarized in Table 3.

## 4. Discussion

### 4.1. Hypoechoic Characteristics of MTrPs

The results of stage 1 reveal that the images obtained from the ultrasonography of MTrPs, which was performed after an indwelling needle was inserted upon the detection of EPN, all indicated hypoechogenicity. Therefore, the MTrP sites of the masseter muscles of the rats appeared hypoechoic in ultrasound images. When an ultrasound detects tissue interfaces with varying densities, part of the emitted energy is reflected. The intensity of a grayscale image is determined by the amount of ultrasonic energy that is reflected. When more energy is reflected, an ultrasound image exhibits more hyperechogenicity; that is, bright spots appear whiter in the image. The strength of an ultrasound penetration or reflection is related to the ratio of acoustic impedance between tissue interfaces. A greater difference in acoustic impedance indicates a greater amount of reflected ultrasonic energy [26].

The integrated hypothesis posits that acetylcholine concentration considerably increases at the neuromuscular junction of an MTrP site [27]. The sustained contraction of muscle fibers causes local ischemia, hypoxia [28,29], and vasocontraction [30]. The ultrasound images of muscle injuries characterized by delayed onset muscle soreness revealed hyperechoic hematomas [31]. Therefore, the ischemia of MTrPs may explain the decreasing difference between their acoustic impedance and those of the surrounding tissues, which results in MTrP images exhibiting hypoechogenicity. In addition, the previous sono-histological research revealed that the hypoechogenicity of the MTrP could partially be explained by the fluids entrapped inside the micro-cracks and fissurations of the intercellular scaffold of the muscle [32].

### 4.2. Performance of Ultrasonography

In addition to revealing that the ultrasound images of MTrPs indicate hypoechogenicity, this study verified the performance of ultrasonography for detecting MTrPs. The verification process primarily occurred in stage 2, during which ultrasonographic MTrP images were captured and then a needle was inserted into an MTrP site (as identified under ultrasonic guidance) in order to detect EPN. In stage 1, ultrasonography was performed to verify the existence of MTrPs; this was achieved using the strong echo image of a needle tip that was generated during the search for MTrPs at a 100% sensitivity level. This same level of sensitivity was also observed in stage 2, indicating that ultrasound images exhibit hypoechogenicity when an MTrP is present in the muscle. This high-sensitivity feature is helpful for clinical interventions (e.g., guided dry needle therapy and injection), which require precise information on the location of an MTrP. Consequently, the ultrasound guidance is significant not only to locate the MTrP but also to accurately reach it with the needle while avoiding iatrogenic injuries to the surrounding tissues. The specificity (i.e., the proportion of correctly identified actual negatives) in stages 1 and 2 was 93.3%, and 40%, respectively. Therefore, when no MTrP is present in an observed muscle, the ultrasound images of the muscle may be misinterpreted, producing false-positive results. Accordingly, an ultrasonography-based MTrP evaluation should be combined with observations of clinical symptoms and palpation, all of which contribute to the location of MTrPs. Regarding the false-positive results, however, the scholars proposed that the hypoechogenicity also related to the presence of muscular fascicles with different spatial orientations. Some authors described an “oscillatory” technique with the ultrasound transducer in order to reduce the acoustic artifacts, and thus reduce the false-positive results [32]. Therefore, the technical aspect is very important for clinical application and a cross-match between clinical symptoms and sonographic findings should always be performed as a daily practice. Additionally, compressing the hypoechoic nodular findings with the ultrasound transducer to assess if the sonographically detected nodule is painful or not (i.e., sono-palpation) may provide better specificity [33].

A positive predictive value reflects the proportion of true positive results obtained in diagnostic tests. In the context of this study, it reflects the positive results obtained through the ultrasonic scan performed in stage 2, which revealed that the proportion of actual MTrPs was 89.2%. In stages 1 and 2, a negative predictive value of 100% indicates that when no MTrP is detected through ultrasonography, an evaluator can confidently determine that no MTrP is present in the ultrasound field of view of the examined muscle. The positive likelihood ratio of only 1.67 was obtained in stage 2, indicating that ultrasonography can be used to obtain hypoechoic images and that the probability of the presence of MTrP is only 1.67 times greater than that of the absence of an MRrP. Therefore, ultrasonography should be combined with an analysis of clinical symptoms and a palpation examination, both of which can help confirm the existence of MTrPs. A negative likelihood ratio of zero was obtained in this study, indicating that the use of ultrasound images to determine the presence of MTrPs produces robust results.

An evaluation of the use of ultrasonography in combination with EPN detection revealed that ultrasonography achieved an accuracy of up to 89.9% for detecting MTrPs. In addition, the precision (i.e., a reliability-related measure of the closeness of two or more measurements to each other) of MTrP detection through ultrasonography reached 89.2%. The results of this study indicate that ultrasonography is a highly accurate and precise method for detecting MTrPs.

### 4.3. Clinical Application

Numerous studies have proposed various treatments for MTrP, including laser therapy [22,34], dry needle therapy [35,36], extracorporeal shock wave therapy [37], and manual therapy [38]. The primary method for measuring the outcome of MTrP treatments is the pain index; however, pain index data are regarded as subjective data. If ultrasound images can be effectively used to detect MTrPs and can be combined with image analysis technology to obtain morphological data (e.g., MTrP area and thickness), this method can be applied to evaluate the outcomes of MTrP treatments. It is important for a clinician to put forward objective evidence that is of clinical efficacy, not only for professional recognition, but also for patient protection, which will help improve the quality of the treatment.

### 4.4. Limitations and Directions for Future Research

Ultrasonography is widely used in the clinical diagnosis of musculoskeletal diseases, including muscle strains, ligaments or tendon sprains, and muscle tumors. If it is validated, the use of 2D ultrasound images for MTrP detection provides a cost- and time-efficient method for the clinical diagnosis of MTrPs. In this study, rats were examined to detect the MTrPs of their chewing muscles, and the results obtained from rats cannot be directly extrapolated to humans because the active or latent status of MTrPs could not be determined in this study. That is, the effect of the active or latent status of MTrPs on the performance of ultrasonography for MTrP detection remains unclear. Therefore, to achieve a comprehensive understanding of the performance of ultrasonography for MTrP detection, studies should enroll human participants and apply the procedures used in this study.

In the present study, only the 2D ultrasound image was used for MTrP detection. Therefore, the other function of ultrasound, such as color/power Doppler or elastography, to identify MTrP are worthy of further investigation.

## 5. Conclusions

In this study, the EPN spontaneously generated by MTrPs was used to verify the characteristics of MTrP ultrasound images, which were revealed to be hypoechoic. The performance of ultrasonography for MTrP detection was satisfactory; specifically, a sensitivity level of 100% and accuracy and precision levels of >89% were achieved. We also demonstrated that ultrasonography is an excellent tool for diagnosing MTrPs; however, in this study, a specificity of only 40% was achieved, indicating that ultrasonography tends to produce false-positive results when an observed muscle has no MTrP. The use of ultrasonography to distinguish MTrPs from surrounding tissue should be further developed to facilitate the determination of MTrP morphology and provide a clinical framework for assessing the effects of MTrP treatments.

## Figures and Tables

**Figure 1 sensors-24-00718-f001:**
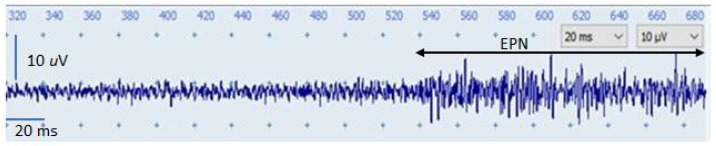
Continuous EPN tracing and visualization of its changes on an EMG screen (EPN: endplate noise; EMG: electromyography).

**Figure 2 sensors-24-00718-f002:**
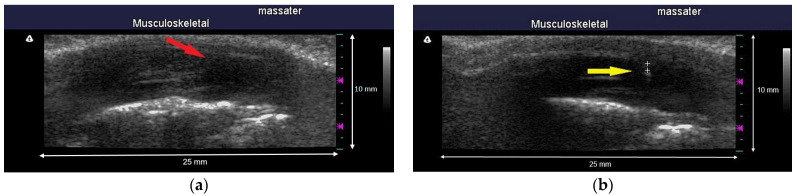
(**a**) The myofascial trigger points are visualized as a hypoechoic region which the red arrow points to with an elliptical appearance in ultrasound imaging. (**b**) The ultrasound image shows both the needle tip and the myofascial trigger points as indicated by the yellow arrow in the picture. The location of the needle tip indicates the detection of continuous EPN; any nodule identified at this site is hypoechoic.

**Table 1 sensors-24-00718-t001:** Crosstab of data obtained from stage 1.

		EPN		
		Positive	Negative	Total
US Image	Positive	148	2	150
	Negative	0	28	28
Total		148	30	178

**Table 2 sensors-24-00718-t002:** Crosstab of data obtained from stage 2.

		US Image		
		Positive	Negative	Total
EPN	Positive	148	0	148
	Negative	18	12	30
Total		166	12	178

**Table 3 sensors-24-00718-t003:** Performance of ultrasonography for assessing myofascial trigger points.

		Ss	Sc	PPV	NPV	PLR	NLR	A	P
Stage	1	100%	93.3%	98.7%	100%	15	0	98.9%	98.7%
	2	100%	40%	89.2%	100%	1.67	0	89.9%	89.2%

Ss: Sensitivity; Sc: Specificity; PPV: Positive predictive value; NPV: Negative predictive value; PLR: Positive likelihood ratio; NLR: Negative likelihood ratio; A: Accuracy; P: Precision.

## Data Availability

The data that support the findings of this study are available from the corresponding author upon request.

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
