# Peer review of "Assessment of the Performance of Ultrasonography for Detecting Myofascial Trigger Points"

_sensors, 2024, doi:10.3390/s24030718_

Round 1
Reviewer 1 Report
Comments and Suggestions for Authors
This study aimed to determine the echogenicity of myofascial trigger points (MTrPs) using ultrasound imaging and a rat model including the masseter muscle. Although interesting, I have some concerns that need to be addressed. See below.
1) Abstract: “whether they are hyperechoic or hypoechoic in ultrasound images is still unclear” – Please describe why this is useful to know. How can we use this information?
2) In the abstract, it is not clear how many recordings have been considered. It seems like there is only two recordings, one for stage 1 and another for stage 2. Please clarify in the abstract for the readers
3) Please double-check the styling, I found a couple of missed dots in some sentences. For example, in the last sentence in the abstract and the last sentence in the conclusions.
4) Methods: I note that you have included Sprague-Dawley rats with varying weights (250-300 g), suggesting more than one rat is included. However, it does not state how many. So, how many rats were included in the study?
5) What was the pitch of the ultrasound transducer? The size of the transducer must have been a lot bigger than the masseter muscle of a rat (although it is a male).
6) In Figure 2, the scale is missing, so it is difficult to understand the field of view sizes. Moreover, I presume it is difficult to say these are MTrPs based on their hypoechoic regions since many things can be hypoechoic. How can you be sure it is not something other than an MTrP?
7) Please clearly describe the number of recordings and animals used in Stage 1 and 2.
8) “The parameter data obtained in this study exhibited both ‘positive’ and ‘negative’.” – I do not understand this sentence. Please elaborate
9) It is unclear how the number of rats and recordings is associated with the numbers in Tables 1 and 2. And what is the total amount referring to?
10) “To the best of our knowledge, the present study is the first to investigate the echogenicity of MTrPs.” – I do not think this is true. For instance, ref 11 in the manuscript studies MTrPs and clearly shows hypoechoic regions related to these. So, at best, does it confirm previous findings?
Author Response
Thank you very much for taking the time to review this manuscript. Please see the attachment.

Reviewer 2 Report
Comments and Suggestions for Authors
It was a great pleasure for me to review the present research about the performance of ultrasonography for detecting myofascial trigger points. Considering the two different diagnostic techniques used by the authors to cross-match the US and EMG findings, it can be considered very interesting for the readers of this journal. Likewise, some revisions are necessary before the eventual publication.
Figure 2
I suggest the authors specify in the corresponding legend the meaning of the red arrow, yellow arrow, and the calipers visible in the images. Please, revise it.
Table 1 & 2
I would replace "Image" with "US Image".
Table 3
I suggest reformatting the table because several words are "interrupted" on the upper side of the table.
Discussion
The nodular hypoechogenicity described by the authors in the present study can be partially explained by the content of water inside the MTrP. Indeed, previous sono-histological research demonstrated that the reduced echogenicity of the MTrP compared to the surrounding "normal" muscle tissue can be partially explained by the fluids entrapped inside the micro-cracks and fissurations of the intercellular scaffold of the muscle (i.e., the endomysium). At this level, a high concentration of glycosaminoglycans has been demonstrated, which are highly hygroscopic and prone to absorbing fluids and nociceptive substances. This cross-match between histological and sonographic features of the MTrP is pivotal and should be briefly described in the text by the authors. For a detailed description of this topic, please refer to Am J Phys Med Rehabil. 2023 Jan 1;102(1):92-97. doi: 10.1097/PHM.0000000000001975.
The authors have mentioned the pivotal role of ultrasound to accurately locate the MTrP and plan tailored interventions (e.g., dry needling and injection). I suggest specifying that ultrasound guidance is very useful not "only" to identify the MTrP but also to accurately target it with the needle avoiding iatrogenic injuries to the surrounding neurovascular elements. In this sense, accuracy but also safety can be optimized by using ultrasound imaging in the management of patients with MTrP.
Limitations
I suggest the authors better describe in the text all the different limitations of the present study:
- the absence of color/power Doppler assessment of the MTrP
- the absence of advanced software to assess the "stiffness" of the MTrP such as the elastography
- control group?
- etc.
"We also demonstrated that ultrasonography is an excellent tool for diagnosing MTrPs; however, in this study, a specificity of only 40% was achieved, indicating that ultrasonography tends to produce false-positive results when an observed muscle has no MTrP."
This technical aspect is very important and should be better described in the text. Indeed, considering the high anatomical variability of the muscular architecture, focal hypoechogenicity can also be related to the presence of muscular fascicles with different spatial orientations. Some authors in the pertinent literature have also described an "oscillatory" technique with the ultrasound transducer to reduce the acoustic artifacts, increase the diagnostic accuracy, and reduce the false-positive results (i.e., fake hypoechoic nodule). In this sense, a cross-match between clinical and sonographic findings should be always performed in daily practice. For a detailed description of this technical topic, please refer to Am J Phys Med Rehabil. 2023 Jan 1;102(1):92-97. doi: 10.1097/PHM.0000000000001975.
Lastly, in the clinical practice is possible to promptly compress with the ultrasound transducer the hypoechoic nodular finding to assess if the sonographically-detected nodule is painful or not (i.e., sono-palpation). In this sense, the ultrasound examination in real patients (and not animals) may provide a better specificity.
Comments on the Quality of English LanguageMinor editing of the English language required
Author Response

(The authors gave the same response as above.)

Round 2
Reviewer 1 Report
Comments and Suggestions for Authors
Thank you for addressing my comments, which improved the manuscript considerably. I have a final question that I find important for readers to clarify. How were the 178 recordings distributed on the 45 rats? On average, there are about 4 recordings per rat. Did the recordings on the same rat quantify the very same MTrPs?
Author Response
Thanks for your comment. A total of 45 rats were included. Bilateral masseter muscles of each rat were measured twice in each stage. However, 2 recordings were discarded due to poor quality. So there were 178 recordings in each stage. (Page 4, ‘Results’, Line 168-170)
The bilateral masseter muscles of each rat were measured at the same time. After a few days of rest, the second measurement was taken. The recording was mainly to detect whether the hypoechoic image and EPN of MTrP appear. Whether quantitative measurements such as the size of MTrP were performed on the very same MTrP was not the purpose of this experiment.
Reviewer 2 Report
Comments and Suggestions for Authors
The revised manuscript can be accepted for publication.
Author Response
We appreciate the helpful comments from the reviewer that provided insight and suggestions for improving the interpretation of the paper. Thank you for the review of our manuscript.